# Citrox Improves the Quality and Shelf Life of Chicken Fillets Packed under Vacuum and Protects against Some Foodborne Pathogens

**DOI:** 10.3390/ani9121062

**Published:** 2019-12-02

**Authors:** Hany Mohamed Yehia, Manal Fawzy Elkhadragy, Wafa Abdullah Al-Megrin, Abdulrahman Hamad Al-Masoud

**Affiliations:** 1Department of Food Science and Nutrition, College of Food and Agriculture Science, King Saud University, Riyadh 11451, Saudi Arabia; 2Department of Food Science and Nutrition, Faculty of Home Economics, Helwan University, Cairo 11221, Egypt; 3Biology Department, Faculty of Science, Princess Nourah bint Abdulrahman University, Riyadh 11671, Saudi ArabiaWafa.megren@gmail.com (W.A.A.-M.); 4Zoology Department, Faculty of Science, Helwan University, Cairo 11790, Egypt

**Keywords:** citrox (1–2%), MRSA, TVC, TVBN, total coliform, pH

## Abstract

Natural antibacterial agents such as citrox are effective against many foodborne pathogens and foods contaminated with bacteria. We studied the antimicrobial effects of citrox solutions (1% and 2%) on the total viable counts of methicillin-resistant *Staphylococcus aureus* (MRSA) in chicken meat fillets. The total coliform group counts found in the chicken samples were also determined. The samples were treated with *S. aureus* at a concentration of 10^6^ colony-forming units (cfu)/g of meat and vacuum-packed (VP) at 4 °C for 3, 6, 9, 12, 15, 18, and 21 days. We also studied the effect of citrox on the total volatile basic nitrogen (TVBN) content and pH changes during the storage period of the meat samples. The results revealed that citrox inhibited the growth of MRSA in the chicken fillets. The total viable counts of MRSA decreased after treatment with 2% citrox in all treated samples that were stored at 4 °C by approximately 2 log units compared with the samples inoculated with *S. aureus* (Chicken-Staph groups) after 3, 6, 9, and 12 days of storage, and by approximately 1 log unit compared with the control samples treated with salt (Chicken-Salt groups) after 3, 6, and 9 days of storage. TVBN was reduced in the Chicken-Citrox-treated samples stored at 4 °C compared with the Chicken-Staph- and Chicken-Salt-treated samples. The results indicated that citrox is effective in reducing the total counts of MRSA and in improving the quality of chicken during the first three days of storage by reducing the number of bacteria by 1 log unit and extending the shelf life of chicken.

## 1. Introduction

Methicillin-resistant *Staphylococcus aureus* (MRSA) was identified in 1962 and, together with certain species of *Enterococcus*, is currently considered a global pandemic threat [1,2]. Methicillin-resistant *S. aureus* is classified into three groups, of which healthcare-associated methicillin-resistant *S. aureus* (HA-MRSA) is considered a major causative agent of chronic diseases and is present in catheters among other places. After two decades from its discovery, the first case of acquired community-associated *S. aureus* MRSA (CA-MRSA) was reported in many countries [3], as was livestock-associated *S. aureus* (LA-MRSA). LA-MRSA has also been reported to be associated with companion animals [4,5,6,7]. The HA-MRSA and CA-MRSA infections that generally affect humans are not involved in livestock infections. However, LA-MRSA may affect humans, especially in the case of occupational contact with livestock [8]. Although many foods containing CA-MRSA, LA-MRSA, and HA-MRSA have been documented, it is not clear whether MRSA can be classified as a food-borne pathogen. 

MRSA is found in several species of animals, such as pigs [9,10], poultry [4], and cattle [11], and their meat products. De Boer et al., 2009 [12] found MRSA to be present predominantly in turkey (35.3%), followed by chicken (16.0%), veal (15.2%), pork (10.7%), and beef (10.6%). Boost et al. [13] recovered MRSA from 455 fresh and frozen chicken samples and found 31 of them (6.8%) to be positive. Many studies have reported *S. aureus* infections in poultry. Multidrug-resistant *S. aureus* has occurred in US meat and poultry at a rate of 52% [14]. Poultry meat is highly perishable and provides a high nutritive medium for the growth of bacteria and other spoilage and pathogenic microorganisms [15]. The increasing production and global demand for poultry meat has also increased the importance of poultry meat hygiene and safety worldwide [16].

Chemical parameters that are considered to determine the microbial contamination in meats and fish include total volatile basic nitrogen (TVBN) [17,18]. TVBN in dark turkey meat packed under aerobic conditions has been shown to correlate well with microbial growth and the proliferation of spoilage microorganisms [17]. The other parameter associated with microbial growth is the change in pH [18,19]. These two factors are considered important indicators of microbial spoilage in meat and poultry [17,20]. 

The preservatives employed to inhibit the growth of spoilage and pathogenic microorganisms have high acidity. However, consumers consider their use undesirable, and their demand has reduced the levels of such additives in foods. Although the safety of these foods is supposed to be ensured primarily by the low pH of these additives, several pathogens, namely, *Escherichia coli* O157:H7, Listeria *monocytogenes*, and *Salmonella* spp., have been reported to survive or even grow in these foods [21]. Usually, the carriers of these pathogens are the raw ingredients, as well as any contaminations from the processing environment and packaging operations.

The present study aimed to evaluate the effect of different concentrations of citrox (1% and 2%) on MRSA in vitro. Moreover, its effects on the survival and elimination of bacterial growth to prolong the shelf life of vacuum-packaged (VP) chicken fillets stored at 4 °C for 0, 3, 6, 9, 12, 15, 18, and 21 days were studied. The study also aimed to determine the changes in TVBN and pH during storage.

## 2. Materials and Methods

### 2.1. Chicken Fillet Sample Preparation

Cooled chicken fillet samples (skinless and boneless), weighing approximately 25 g, were collected from poultry retail markets located in Riyadh, Saudi Arabia. The samples were incubated at 2 °C for further experiments.

### 2.2. Methicillin-Resistant S. aureus

MRSA obtained from the laboratory of food microbiology, College of Food Science, King Saud University, was used as the inoculum (10^6^ colony-forming units (cfu)/mL). For activation of the strain, 1 mL of overnight culture was added to 9 mL of brain heart infusion broth (HiMedia, M210-500G) and incubated at 37 °C for 24 h. Subsequently, the samples were centrifuged to collect the sediments, which were washed twice with saline solution (0.85% NaCl) before the preparation of the final solutions.

### 2.3. Citrox Solution 

The citrox solution was prepared in the laboratory by mixing 18 g of citric acid, 18 g of malic acid, and 5 g of ascorbic acid in 100 mL of water. The pH of the solution was adjusted to ~2.7. The solution was yellow in color. The citrox solution was subsequently diluted to 1% and 2% and sterilized at 121 °C for 15–20 min along with the addition of 0.85% NaCl. 

### 2.4. Chicken Fillet Samples: Inoculation, Treatment, and Packaging 

Five hundred and twelve chicken fillet samples (weighing approximately 25 g each) were divided into four main groups: A, B, C, and D (each containing 128 samples).

Group A, Chicken-Salt: The samples were soaked in a tray containing 1 L of salt solution (0.85% NaCl) and left for 2 min. This was used as the negative control. 

Group B, Chicken-Salt-Staph: The samples were soaked in 1 L of saline solution and then drained through a sterilized sieve. Subsequently, the samples were soaked in the MRSA inoculum (10^6^ log cfu/mL) and left for 2 min. This was used as the positive control. 

Groups C and D, Chicken-Salt-Staph-Citrox 1% and 2%, respectively: The samples were soaked in 1 L of saline solution for 2 min followed by soaking in MRSA for another 2 min. Subsequently, 1 L of citrox solution (final concentration of 1% or 2% in the salt solution for groups C and D, respectively) was added after draining the samples through a sieve. All samples were left on the sieve to dry before packing them in transparent polyethylene pouches (low density), sealing under vacuum (Plas Vac 20, Komet, Germany), and finally storing in cooled incubators at 4 °C for different intervals of time (0, 3, 6, 9, 12, 15, 18, or 21 days). These samples were used for the determinations of TVBN, pH, and sensory tests.

### 2.5. Activity of Citrox

The effect of citrox on MRSA was evaluated through an antimicrobial activity technique by inoculating one colony of *S. aureus* in brain heart infusion broth (HIMEDIA, M210) at 37 °C for 24 h. A total of 100 µL of culture (10^6^ cfu/mL) was inoculated into brain heart infusion medium (Oxoid, CM375) using the agar well diffusion method. Then, a hole with a diameter of 6 mm was punched aseptically with a sterile cork borer, and two different volumes (50 mL and 100 mL) of 1% and 2% citrox solution were introduced into each well. For comparison of its effects on MRSA, 1% chitosan was mixed with the same volume of citrox 1 and 2%. The plates were then incubated at 37 °C for 24 h, and the zone of inhibition was observed.

### 2.6. Microbiological Analysis

Chicken fillet samples (10 g) were aseptically transferred to 90 mL of 0.1% peptone water (Oxoid, CM 0009) in a stomacher bag (Seward Ltd., London, UK), and the mixture was homogenized for 1 min at room temperature. For enumeration of methicillin-resistant *S. aureus*, 0.1 mL and 0.01 mL samples of the dilution (1/10) were poured into the brain heart infusion medium (Oxoid, CM375) and incubated at 37 °C for 24–28 h. 

### 2.7. Chemical Analysis

#### 2.7.1. Quantitative Determination of TVBN

For the determination of the total volatile base nitrogen, the magnesium oxide method was used as described previously [22]. Briefly, 5 g of sample was added to a heating flask containing 300 mL of distilled water, 2 g of magnesium oxide, and anti-foaming granules. In the receiving flask, 25 mL of boric acid (2%), with a few drops of Tashiro’s indicator (1.25 g of methyl red + 0.32 g of methylene blue in one liter of 90% ethanol), was added. The two flasks (heating and receiving) were connected to an evaporator, and the water bath was managed. After 25 min, the distillation was stopped. The content of the receiving flask was titrated to the endpoint using sulfuric acid (0.05 N).

The total volatile nitrogen was then determined as follows:TVBN = (V × N × 100 × 14)/W
where V = volume (mL) of H_2_SO_4_ used for the sample, N = normality of H_2_SO_4_ (0.05 N), and W = weight of the sample in grams.

#### 2.7.2. pH Measurements

To measure the pH, 25 g of each sample was added to 10 mL of distilled water and homogenized for 1 min. The pH measurements were performed using a pH meter (pH 8000-Sargent–Welch, Warner Road, Cleveland, OH, USA) in triplicate.

#### 2.7.3. Color Measurements

The surface color of the chicken samples was measured using the CIEL * a * b * color scales based on the opponent color theory. This theory assumes that the receptors in the human eye perceive color as the following pairs of opposites. L scales (0–50): light vs. dark, where a low number indicates dark, and a high number indicates light. A scale: red vs. green, where a positive number indicates red, and a negative number indicates green. B scale: yellow vs. blue, where a positive number indicates yellow, and a negative number indicates blue. The samples were evaluated during the storage period using a colorimeter (CR-300 spectrophotometer, Minolta Inc., Tokyo, Japan), which was calibrated against black and white reference tiles. Measurements of the L *, a *, and b * values from each treatment were recorded at three different locations, each with three replications.

#### 2.7.4. Sensory Panel Evaluations

Chicken fillet samples were cooked using a microwave oven (Sanyo, Model: EM-G1299V, Jiangsu, China) on high power for 10 min. A panel of five experienced food scientists (technicians, staff member, and postgraduate students) were selected to evaluate the sensory attributes of the cooked chicken. All the panelists had previously participated in training sessions to become familiar with the sensory characteristics of cooked chicken. The taste, odor, and appearance of the cooked chicken fillets were used as the testing parameters to evaluate the samples. The acceptability of odor and taste was estimated using a scale ranging from 0 to 9. The panel was asked to indicate whether the products had acceptable or unacceptable (deviating) taste, flavor, and odor. A 9-point hedonic scale was used (9 = like extremely, 5 = like moderately, 1 = dislike extremely) in this study.

#### 2.7.5. Statistical Analysis

All the results are presented as the mean ± standard deviation (M ± SD). Analysis of variance (ANOVA) was used to determine the differences between the groups and periods of storage in a completely randomized factorial design [23]. When a significant main effect was detected, the means were separated with Duncan’s multiple test. Differences between groups with *p* ≤ 0.05 were considered significant.

## 3. Results and Discussion

Inoculation with 1% and 2% citrox solutions inhibited the growth of MRSA. However, the zone of inhibition was observed to be larger with 2% citrox. Since both concentrations were effective, we decided to treat the chicken fillets with both (1% and 2%) to limit or stop the growth of MRSA in chicken fillet samples during the storage period. Chitosan mixed with citrox at both concentrations (1% and 2%) did not appear to have any effect on the growth of MRSA, so there was no need to use it in our experiments.

The total viable count (TVC), which estimates the concentration of microorganisms in a sample, is commonly used as a microbiological parameter. It determines the hygiene status and shelf life of meat and meat products. The total viable counts in chicken fillets during the 21 days of storage are presented in Figure 1. The meat samples treated with the MRSA solution (positive control) recorded the highest total viable count of 8.200 log cfu/g after 3 days and 8.733 log cfu/g after 9 days of storage, exceeding the maximum set limit. However, in the chicken samples treated with 0.85% NaCl (negative control), the total count increased gradually from 7.200 log cfu/g on the 12th day and reached 8.433 log cfu/g on the 21st day of storage. The group C samples stored at 4 °C revealed a stable TVC of 6.733, while the TVC of the D group was 6.567 log cfu/g after 21 days of storage, and both groups C and D did not exceed the permissible limit. The decreased in TVC of Chicken-Staph samples after 10 days of storage may be referred to the increment in microbial metabolites, so the growth curve entered the decline phase. The samples retained their temperament and were devoid of the corruption found in the treated samples throughout the storage period. The TVC of the samples treated with 2% citrox revealed a reduction by 2.5 and 2.0 log units on the 9th day compared to the negative and positive controls, indicating the significant effect of citrox on the bacteria. Moreover, the temperature of domestic and retail storage refrigerators, which are considered critical points of the cold chain, often range from −1 to 15 °C [24,25]. Temperature is one of the most important factors affecting microbial growth [26,27,28]. In fact, a microbial community changes according to variations in the temperature [28,29].

TVBN, a product of microbial amino acid decarboxylase activity, is used to estimate the shelf life of chicken meat. In the present study, as shown in Figure 2, after 21 days of storage, the TVBN values in chicken treated with 1% and 2% citrox were 48.067 mg/100 g and 35.933 mg/100 g at 4 °C, respectively. These values were significantly lower compared to those of the chicken breast fillets treated with salt, as a negative control, (99.867 mg/100 g) and *S. aureus* (MRSA), as a positive control (62.533 mg/100 g). The TVBN values of citrox-treated chicken did not exceed the recommended limit of 60 mg/100 g (NF V 01-003, 2004) during the entire storage period of 21 days, and the treatment with 2% citrox was better than that with 1% citrox. The TVBN in all groups increased over the study period, but that of the Chicken-Staph-Citrox group treated with citrox 1% an 2% did not exceed 60 mg/100 g over the entire study period. Storing chicken breast fillets at 4 °C under vacuum was more effective against MRSA, which led to a reduction in the loss of chicken meat quality and improved its safety. The chemical method used for determining the spoilage of chicken and chicken products was the determination of TVBN, which serves as an indicator of its quality and safety. Generally, consumers throughout the world do not prefer foods treated with various chemical additives; instead, the demand for natural products being used as preservatives is increasing. Citrox is composed of ingredients of plant origin (citric, ascorbic, and malic acids) and can be used as an alternative preservative because it is effective in the presence of organic matter, breaks down biofilms, extends shelf life, reduces pathogenic attacks, can be applied directly to food as an additive, and conforms to the European Suspension Test (BS EN 1276). Several studies have also reported TVBN to be an indicator of the quantity of biogenic amines, which are produced during microbiological contamination of foods [30,31,32].

### 3.1. pH Value

During all the days of the experiment, no statistical variance in the pH value was detected among the four groups (Figure 3). The initial pH of the chicken samples treated with salt was 6.010, and this value reached 6.650 after 21 days of storage. Chicken samples treated with 1% and 2% citrox had an initial pH of 6.00, which reached 6.433 and 6.400, respectively, on the 21st day. Highly perishable food that is stored under vacuum, such as chicken, shows an increase in pH because of the number of microorganisms that cause spoilage. The relative increase in the pH of the salt-treated chicken compared to that of the MRSA-treated chicken could be attributed to this fact. The proteolytic activity of mixed microorganisms in a basic material leads to a greater increase in pH than that of one type of bacterium. Chicken treated with 1% and 2% citrox solutions also showed an increase in the pH, but this increase was lower than that for the salt and MRSA treatments. Rio et al. [33] reported that dipping chicken meat in citric acid significantly decreased the pH after marination. The reduction of the pH value of meat products is considered to influence many factors during the storage period, such as extension of the storage period, stability of the water-binding capacity and texture, and loss of redness [34].

### 3.2. Color Value

The four treatment groups of chicken breast fillets observed in the present study were examined on the basis of color differentials, as explained in Table 1. Significantly higher L * values were observed for groups C and D treated with 1% and 2% citrox solutions compared to both the negative and the positive controls on day 9 of storage. No significant changes in the L * value were observed between the negative control and the positive control. A significant increase in the a * value was observed after 9 days of storage in the sample treated with Staph (group B) compared to the remaining three groups.

Generally, the reduction in the red color of the chicken breast fillet samples during the storage period was due to the correlation between lipids and pigment oxidation in the meat. It could be due to the binding of the heme iron of myoglobin with citric acid or the prevention of the formation of pink pigments by acidification [35].

However, a significant increase in the b * value was detected in samples treated only with Staph compared with the other three treatments. The effects of chilling and using various concentrations of the acidic citrox led to a change in the skin color of the chicken breast fillets.

The organic acid preservatives present in the citrox solution (citric, malic, and ascorbic acids) increased the lightness and decreased the redness and yellowness values. These results are in agreement with those of Bilgili et al. [36], who reported that propionic acid had little effect on the lightness and redness values but significantly decreased the yellowness values. Similarly, the processing conditions, such as scalding temperature and storage period [37,38], pH [38], and immersion chilling [39], have been shown to affect broiler skin color. Some studies have reported that the use of an appropriate amount of lactic acid is important to determine the effect on the color of fresh and cooked meat [40]. Lactic acid concentrations of 1.2% and 1.5% caused color deterioration in beef samples during display [40]. Dipping meat in a citric acid (CA) solution has been proven to prevent redness in products conferred by the sous-vide process [41]. The increase in CA concentration had a positive influence on the reduction of the pink color by inducing the thermal denaturation of myoglobin during refrigerated storage.

### 3.3. Sensory Evaluation

A sensory analysis test of the chicken fillet samples revealed that the sample injected with MRSA resulted in lower sensory scores for all parameters compared to the control (Table 2). The addition of 1% and 2% citrox-treated samples acquired moderate sensory scores for color, odor, and flavor but lower scores for taste and tenderness. The use of citrox containing a combination of flavorings and antioxidants could improve the nutritional and sensory attributes of meat. However, further studies are required to obtain such benefits without compromising microbial safety.

The addition of a citrus extract at a concentration of 0.1 mL/100 g, when used singly or in combination with an oxygen absorber, reduced the TVC of aerobically packaged ground chicken meat by 0.5 and 1.5 log cfu/g, respectively [42]. The combination of a citric extract and chitosan (CH) was added to turkey meat to improve its taste and odor [43], and the current results are in agreement with the findings of the authors who reported that chitosan, applied singly or in combination with oregano essential oil (EO), did not negatively influence the taste of chicken breast meat [44]. Both chitosan and citrox were sensorially acceptable when added to turkey samples, with chitosan characterized by spicy, fruity, and oriental flavors, and citrox characterized by a citrus-like flavor. Moreover, the addition of both a citrus extract and chitosan may provide the possibility of new flavors and options for poultry products. However, further sensory tests are needed to examine this possibility [45].

## 4. Conclusions

We have concluded that the effects of citrox on the shelf life of chicken treated with MRSA, in which limited growth was observed in the samples. Treatment of chicken fillets with 2% citrox extended the shelf life by >21 days at 4°C. Citrox at 2% concentration was reduced the total counts of methicillin-resistant *Staphylococcus aureus* (MRSA) by approximately 3 log units and improved also the sensory characteristics of cooked chicken.

## Figures and Tables

**Figure 1 animals-09-01062-f001:**
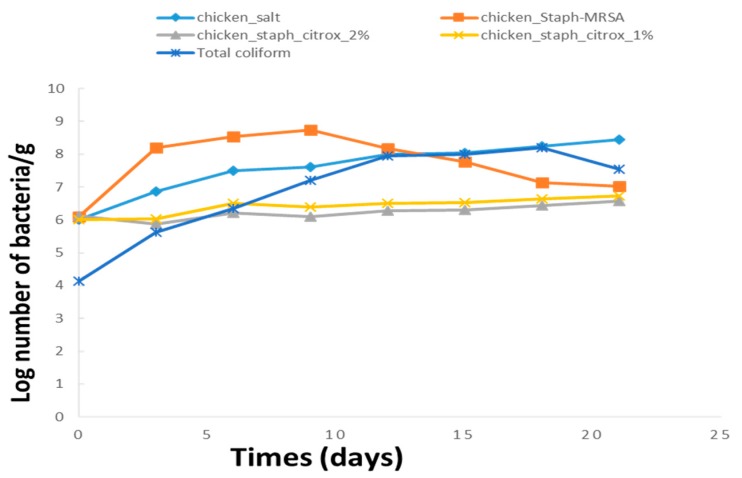
Total viable counts of methicillin-resistant *Staphylococcus aureus* (MRSA) and total viable counts of coliform during storage at 4 °C.

**Figure 2 animals-09-01062-f002:**
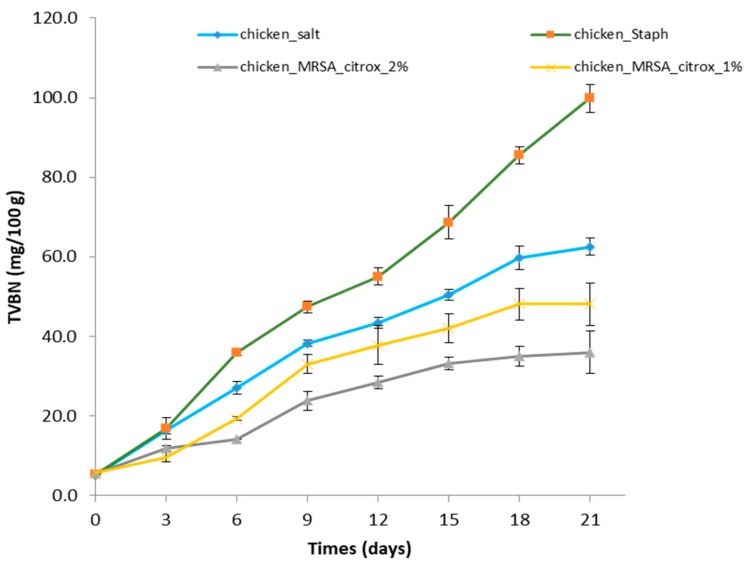
Total volatile base nitrogen of chicken fillet samples during storage at 4 °C.

**Figure 3 animals-09-01062-f003:**
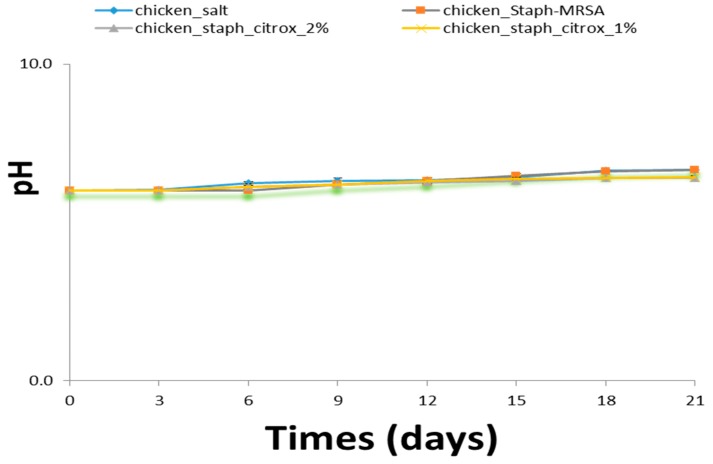
pH values of chicken fillet samples during storage at 4 °C.

**Table 1 animals-09-01062-t001:** Hunter L *, a *, and b * color values of chicken fillet samples treated with 1% and 2% citrox and stored at 4 °C for 21 days.

Hunter Color	Sample Treatment	Storage Period (Days)								
		0	3	6	9	12	15	18	21	Mean
L *	Chicken-Salt (Control)	50.56	51.02	51.53	51.89	52.36	49.43	48.55	47.53	50.36 ^a^
	Chicken-Staph*-MRSA*	50.62	52.22	53.66	52.67	53.16	49.72	47.47	47.11	50.83 ^b^
	Chicken-Staph-Citrox 1%	50.50	52.67	54.65	67.13	55.60	51.78	50.26	50.00	54.10 ^d^
	Chicken-Staph-Citrox 2%	49.37	51.41	52.46	65.33	53.09	49.61	48.41	48.14	52.23 ^c^
	Mean	50.30 ^d^	52.00 ^e^	53.18 ^f^	59.35 ^h^	53.551 ^g^	50.133 ^c^	4877 ^b^	48.20 ^a^	-
a *	Chicken-salt (Control)	1.500	3.690	4.050	4.025	3.555	1.100	0.933	0.835	2.46 ^b^
	Chicken-Staph*-MRSA*	0.890	2.185	3.310	6.505	3.445	3.250	3.210	3.045	3.23 ^a^
	Chicken-Staph-Citrox 1%	0.950	1.165	2.345	5.385	0.555	0.260	0.125	0.110	1.36 ^d^
	Chicken-Staph-Citrox 2%	1.817	2.070	3.877	6.603	1.767	1.437	1.050	0.810	2.43 ^c^
	Mean	1.90 ^e^	2.38 ^c^	3.40 ^b^	5.63 ^a^	2.33 ^c^	1.51 ^d^	1.33 ^e^	1.20 ^f^	-
b *	Chicken-salt (Control)	6.380	7.710	7.525	7.460	6.580	6.140	4.485	4.265	6.33 ^b^
	Chicken-Staph*-MRSA*	4.995	5.325	9.680	9.475	9.335	14.730	11.375	11.010	9.50 ^d^
	Chicken-Staph-Citrox 1%	5.350	5.510	5.050	6.000	6.870	8.255	7.280	7.005	6.442 ^c^
	Chicken-Staph-Citrox 2%	5.033	5.020	4.517	5.110	5.760	7.197	6.117	5.853	5.58 ^a^
	Mean	5.40 ^a^	5.90 ^b^	6.70 ^c^	7.01 ^d^	7.14 ^e^	9.10 ^g^	7.31 ^f^	7.03 ^d^	-

L * = lightness; a * = redness; b * = yellowness. Superscripted letters a, b, c, d, e, f, g and h are the means of values in the same row.

**Table 2 animals-09-01062-t002:** Sensory scores of chicken breast meat samples treated with 1% and 2% citrox.

Treatments	Color	Odor	Flavor	Taste	Tenderness	Overall Acceptability
Chicken-Salt (Control)	5.3 ^a^	5.00 ^a^	5.3 ^a^	5.6 ^a^	5.1 ^a^	5.7 ^a^
Chicken-Staph*-MRSA**	3.6 ^b^	2.36 ^b^	2.88 ^b^	2.6 ^b^	4	2.7 ^b^
Chicken-Staph-Citrox 1%	5.2 ^a,b^	4.55 ^a,b^	4.6 ^b^	4.4 ^b^	4.0 ^b^	4.9 ^b^
Chicken-Staph-Citrox 2%	5.1 ^a^^,b^	4.93 ^a,b^	4.9 ^b^	4.65 ^b^	4.1 ^b^	4.00 ^b^
Mean	4.8 ^a^	4.2 ^b^	4.4 ^b^	4.3 ^b^	4.3 ^b^	4.3 ^b^

Nine-point hedonic scale (9 = like extremely, 4-5 = like moderately, 1 = dislike extremely) used for all sensory parameters. ^a,b^ Values with different letters within the same column differ significantly (*p* < 0.05). *** Methicillin resistant *Staphylococcus aureus* (MRSA).

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
