# Peer review of "Citrox Improves the Quality and Shelf Life of Chicken Fillets Packed under Vacuum and Protects against Some Foodborne Pathogens"

_animals, 2019, doi:10.3390/ani9121062_

Round 1

Reviewer 1 Report

AAs did not reply to reviewers comments.

Author Response

Dear Editor of Animals journal

Thank you for your advices about our manuscript animals-642069 and title of :  

 Citrox improves the quality and shelf life of chicken fillets packed under vacuum and protects against some foodborne pathogens

The following is the response to the Reviewer 1 point by point as he recommended

Open Review

English language and style

( ) Extensive editing of English language and style required 
( ) Moderate English changes required 
( ) English language and style are fine/minor spell check required 
(x) I don't feel qualified to judge about the English language and style 

About English editing this is the second times that was done by specific English editing companies.

Yes

Can be improved

Must be improved

Not applicable

Does the introduction provide sufficient background and include all relevant references?

( )

(x)

( )

( )

The introduction was improved again 

Is the research design appropriate?

( )

(x)

( )

( )

The research design  improved

The experiments design

The experiment design reviewed again

Are the methods adequately described?

( )

(x)

( )

( )

Are the results clearly presented?

The results rewritten again

( )

(x)

( )

( )

Are the conclusions supported by the results?

( )

(x)

( )

( )

The conclusion modified

Dear reviewer: please see the file which was in supplemtary materials to sure that I was changed the points you suggested

Dear Editor of Animals journal

Thank you for your advices about our manuscript animals-642069 and title of :  

 Citrox improves the quality and shelf life of chicken fillets packed under vacuum and protects against some foodborne pathogens

The following is the response to the Reviewer 1 point by point as he recommended

Open Review

English language and style

( ) Extensive editing of English language and style required 
( ) Moderate English changes required 
( ) English language and style are fine/minor spell check required 
(x) I don't feel qualified to judge about the English language and style 

About English editing this is the second times that was done by specific English editing companies.

Yes

Can be improved

Must be improved

Not applicable

Does the introduction provide sufficient background and include all relevant references?

( )

(x)

( )

( )

The introduction was improved again 

Is the research design appropriate?

( )

(x)

( )

( )

The research design  improved

The experiments design

The experiment design reviewed again

Are the methods adequately described?

( )

(x)

( )

( )

Are the results clearly presented?

The results rewritten again

( )

(x)

( )

( )

Are the conclusions supported by the results?

( )

(x)

( )

( )

The conclusion modified

Dear reviewer: please see the file which was in supplemtary materials to sure that I was changed the points you suggested

Dear Editor of Animals journal

Thank you for your advices about our manuscript animals-642069 and title of :  

 Citrox improves the quality and shelf life of chicken fillets packed under vacuum and protects against some foodborne pathogens

The following is the response to the Reviewer 1 point by point as he recommended

Open Review

English language and style

( ) Extensive editing of English language and style required 
( ) Moderate English changes required 
( ) English language and style are fine/minor spell check required 
(x) I don't feel qualified to judge about the English language and style 

About English editing this is the second times that was done by specific English editing companies.

Yes

Can be improved

Must be improved

Not applicable

Does the introduction provide sufficient background and include all relevant references?

( )

(x)

( )

( )

The introduction was improved again 

Is the research design appropriate?

( )

(x)

( )

( )

The research design  improved

The experiments design

The experiment design reviewed again

Are the methods adequately described?

( )

(x)

( )

( )

Are the results clearly presented?

The results rewritten again

( )

(x)

( )

( )

Are the conclusions supported by the results?

( )

(x)

( )

( )

The conclusion modified

Dear reviewer: please see the file which was in supplemtary materials to sure that I was changed the points you suggested

Dear Editor of Animals journal

Thank you for your advices about our manuscript animals-642069 and title of :  

 Citrox improves the quality and shelf life of chicken fillets packed under vacuum and protects against some foodborne pathogens

The following is the response to the Reviewer 1 point by point as he recommended

Open Review

English language and style

( ) Extensive editing of English language and style required 
( ) Moderate English changes required 
( ) English language and style are fine/minor spell check required 
(x) I don't feel qualified to judge about the English language and style 

About English editing this is the second times that was done by specific English editing companies.

Yes

Can be improved

Must be improved

Not applicable

Does the introduction provide sufficient background and include all relevant references?

( )

(x)

( )

( )

The introduction was improved again 

Is the research design appropriate?

( )

(x)

( )

( )

The research design  improved

The experiments design

The experiment design reviewed again

Are the methods adequately described?

( )

(x)

( )

( )

Are the results clearly presented?

The results rewritten again

( )

(x)

( )

( )

Are the conclusions supported by the results?

( )

(x)

( )

( )

The conclusion modified

Dear reviewer: please see the file which was in supplemtary materials to sure that I was changed the points you suggested

Dear Editor of Animals journal

Thank you for your advices about our manuscript animals-642069 and title of :  

 Citrox improves the quality and shelf life of chicken fillets packed under vacuum and protects against some foodborne pathogens

The following is the response to the Reviewer 1 point by point as he recommended

Open Review

English language and style

( ) Extensive editing of English language and style required 
( ) Moderate English changes required 
( ) English language and style are fine/minor spell check required 
(x) I don't feel qualified to judge about the English language and style 

About English editing this is the second times that was done by specific English editing companies.

Yes

Can be improved

Must be improved

Not applicable

Does the introduction provide sufficient background and include all relevant references?

( )

(x)

( )

( )

The introduction was improved again 

Is the research design appropriate?

( )

(x)

( )

( )

The research design  improved

The experiments design

The experiment design reviewed again

Are the methods adequately described?

( )

(x)

( )

( )

Are the results clearly presented?

The results rewritten again

( )

(x)

( )

( )

Are the conclusions supported by the results?

( )

(x)

( )

( )

The conclusion modified

Dear reviewer: please see the file which was in supplemtary materials to sure that I was changed the points you suggested

Dear Editor of Animals journal

Thank you for your advices about our manuscript animals-642069 and title of :  

 Citrox improves the quality and shelf life of chicken fillets packed under vacuum and protects against some foodborne pathogens

The following is the response to the Reviewer 1 point by point as he recommended

Open Review

English language and style

( ) Extensive editing of English language and style required 
( ) Moderate English changes required 
( ) English language and style are fine/minor spell check required 
(x) I don't feel qualified to judge about the English language and style 

About English editing this is the second times that was done by specific English editing companies.

Yes

Can be improved

Must be improved

Not applicable

Does the introduction provide sufficient background and include all relevant references?

( )

(x)

( )

( )

The introduction was improved again 

Is the research design appropriate?

( )

(x)

( )

( )

The research design  improved

The experiments design

The experiment design reviewed again

Are the methods adequately described?

( )

(x)

( )

( )

Are the results clearly presented?

The results rewritten again

( )

(x)

( )

( )

Are the conclusions supported by the results?

( )

(x)

( )

( )

The conclusion modified

Dear reviewer: please see the file which was in supplemtary materials to sure that I was changed the points you suggested

Dear Editor of Animals journal

Thank you for your advices about our manuscript animals-642069 and title of :  

 Citrox improves the quality and shelf life of chicken fillets packed under vacuum and protects against some foodborne pathogens

The following is the response to the Reviewer 1 point by point as he recommended

Open Review

English language and style

( ) Extensive editing of English language and style required 
( ) Moderate English changes required 
( ) English language and style are fine/minor spell check required 
(x) I don't feel qualified to judge about the English language and style 

About English editing this is the second times that was done by specific English editing companies.

Yes

Can be improved

Must be improved

Not applicable

Does the introduction provide sufficient background and include all relevant references?

( )

(x)

( )

( )

The introduction was improved again 

Is the research design appropriate?

( )

(x)

( )

( )

The research design  improved

The experiments design

The experiment design reviewed again

Are the methods adequately described?

( )

(x)

( )

( )

Are the results clearly presented?

The results rewritten again

( )

(x)

( )

( )

Are the conclusions supported by the results?

( )

(x)

( )

( )

The conclusion modified

Dear reviewer: please see the file which was in supplemtary materials to sure that I was changed the points you suggested

Dear Editor of Animals journal

Thank you for your advices about our manuscript animals-642069 and title of :  

 Citrox improves the quality and shelf life of chicken fillets packed under vacuum and protects against some foodborne pathogens

The following is the response to the Reviewer 1 point by point as he recommended

Open Review

English language and style

( ) Extensive editing of English language and style required 
( ) Moderate English changes required 
( ) English language and style are fine/minor spell check required 
(x) I don't feel qualified to judge about the English language and style 

About English editing this is the second times that was done by specific English editing companies.

Yes

Can be improved

Must be improved

Not applicable

Does the introduction provide sufficient background and include all relevant references?

( )

(x)

( )

( )

The introduction was improved again 

Is the research design appropriate?

( )

(x)

( )

( )

The research design  improved

The experiments design

The experiment design reviewed again

Are the methods adequately described?

( )

(x)

( )

( )

Are the results clearly presented?

The results rewritten again

( )

(x)

( )

( )

Are the conclusions supported by the results?

( )

(x)

( )

( )

The conclusion modified

Dear reviewer: please see the file which was in supplemtary materials to sure that I was changed the points you suggested

Dear Editor of Animals journal

Thank you for your advices about our manuscript animals-642069 and title of :  

 Citrox improves the quality and shelf life of chicken fillets packed under vacuum and protects against some foodborne pathogens

The following is the response to the Reviewer 1 point by point as he recommended

Open Review

English language and style

( ) Extensive editing of English language and style required 
( ) Moderate English changes required 
( ) English language and style are fine/minor spell check required 
(x) I don't feel qualified to judge about the English language and style 

About English editing this is the second times that was done by specific English editing companies.

Yes

Can be improved

Must be improved

Not applicable

Does the introduction provide sufficient background and include all relevant references?

( )

(x)

( )

( )

The introduction was improved again 

Is the research design appropriate?

( )

(x)

( )

( )

The research design  improved

The experiments design

The experiment design reviewed again

Are the methods adequately described?

( )

(x)

( )

( )

Are the results clearly presented?

The results rewritten again

( )

(x)

( )

( )

Are the conclusions supported by the results?

( )

(x)

( )

( )

The conclusion modified

Dear reviewer: please see the file which was in supplemtary materials to sure that I was changed the points you suggested

Dear Editor of Animals journal

Thank you for your advices about our manuscript animals-642069 and title of :  

 Citrox improves the quality and shelf life of chicken fillets packed under vacuum and protects against some foodborne pathogens

The following is the response to the Reviewer 1 point by point as he recommended

Open Review

English language and style

( ) Extensive editing of English language and style required 
( ) Moderate English changes required 
( ) English language and style are fine/minor spell check required 
(x) I don't feel qualified to judge about the English language and style 

About English editing this is the second times that was done by specific English editing companies.

Yes

Can be improved

Must be improved

Not applicable

Does the introduction provide sufficient background and include all relevant references?

( )

(x)

( )

( )

The introduction was improved again 

Is the research design appropriate?

( )

(x)

( )

( )

The research design  improved

The experiments design

The experiment design reviewed again

Are the methods adequately described?

( )

(x)

( )

( )

Are the results clearly presented?

The results rewritten again

( )

(x)

( )

( )

Are the conclusions supported by the results?

( )

(x)

( )

( )

The conclusion modified

Dear reviewer: please see the file which was in supplemtary materials to sure that I was changed the points you suggested

Dear Editor of Animals journal

Thank you for your advices about our manuscript animals-642069 and title of :  

 Citrox improves the quality and shelf life of chicken fillets packed under vacuum and protects against some foodborne pathogens

The following is the response to the Reviewer 1 point by point as he recommended

Open Review

English language and style

( ) Extensive editing of English language and style required 
( ) Moderate English changes required 
( ) English language and style are fine/minor spell check required 
(x) I don't feel qualified to judge about the English language and style 

About English editing this is the second times that was done by specific English editing companies.

Yes

Can be improved

Must be improved

Not applicable

Does the introduction provide sufficient background and include all relevant references?

( )

(x)

( )

( )

The introduction was improved again 

Is the research design appropriate?

( )

(x)

( )

( )

The research design  improved

The experiments design

The experiment design reviewed again

Are the methods adequately described?

( )

(x)

( )

( )

Are the results clearly presented?

The results rewritten again

( )

(x)

( )

( )

Are the conclusions supported by the results?

( )

(x)

( )

( )

The conclusion modified

Dear reviewer: please see the file which was in supplemtary materials to sure that I was changed the points you suggested

Dear Editor of Animals journal

Thank you for your advices about our manuscript animals-642069 and title of :  

 Citrox improves the quality and shelf life of chicken fillets packed under vacuum and protects against some foodborne pathogens

The following is the response to the Reviewer 1 point by point as he recommended

Open Review

English language and style

( ) Extensive editing of English language and style required 
( ) Moderate English changes required 
( ) English language and style are fine/minor spell check required 
(x) I don't feel qualified to judge about the English language and style 

About English editing this is the second times that was done by specific English editing companies.

Yes

Can be improved

Must be improved

Not applicable

Does the introduction provide sufficient background and include all relevant references?

( )

(x)

( )

( )

The introduction was improved again 

Is the research design appropriate?

( )

(x)

( )

( )

The research design  improved

The experiments design

The experiment design reviewed again

Are the methods adequately described?

( )

(x)

( )

( )

Are the results clearly presented?

The results rewritten again

( )

(x)

( )

( )

Are the conclusions supported by the results?

( )

(x)

( )

( )

The conclusion modified

Dear reviewer: please see the file which was in supplemtary materials to sure that I was changed the points you suggested

Reviewer 2 Report

much improved

Author Response

I was  send the reply in round 1 

Round 2

Reviewer 1 Report

AAs have applied all suggestions. The paper can now be accepetd.

This manuscript is a resubmission of an earlier submission. The following is a list of the peer review reports and author responses from that submission.

Round 1

Reviewer 1 Report

Line 14-15: The samples were treated with 106 cfu/g of meat, vacuum-packed (VP), and 4°C. This statement is not clear, please rewrite this. What sample are you talking about here? Citrox or what? If its citrox, then be specific. Also, change “and 4oC” to at 4oC.

Line 17: …during the storage period of meat samples.

Line 17-24: decreased in growth of MRSA at what concentration of citrox? Please specified.

Line 83-86: Please specify the sources of the citrox contents (manufacturers)

Line 88: five hundred and twelve

149-158: I have concern with this sensory test, apart from the fact that the cooked chicken meat is contaminated with MRSA, there was no information on the blinding of the panel of food scientists tasting the meat. How are we sure these 5 persons are not lab members that are familiar with the whole study and experiment. I am not quite sure of the validity of the result and inference from this section. Were the 5 persons blinded to the treatment groups?

Line 160-164: Write something like Analysis of Variance (ANOVA) was used to determine the difference between the groups and periods of storage in a completely randomized factorial design. Completely randomized factorial design is not a statistical test but a study design.

Result and Discussion

Line 173: were the chicken injected with MRSA or the meat samples treated in MRSA solution?

You are muddling things up here, be explicit as much as possible. State your result succinctly. You are moving from one result to another for example:

You have to describe changes/growth seen at each days of observation. You should move from (line 175) day 3 and 9 for positive control to day 12 of negative control and day 21 (line 177) of group C and D (line 179). You have to be explicit enough.

Line 181-183: at what day of study or were you talking about average reduction across the study days?

Line 186: delete Puhakka, 2010

Line 184-186: You did not measure temperature in your study, why talking about temperature here? Also the sentence does not connect with preceding and subsequent sentences.

Line 186-191: Please rewrite this lines, it does not make sense, the sentences are all over the place and does not marry together.

Figure 1: Please separate the chicken-staph-citrox into 1% and 2% and include it in the figure 1. Why was the TVC for the chicken-staph increased and then declined to almost the level of chicken-staph citrox by day 21? Can you say something about this in the result and discussion? Also, please change the days on the figure 1 from Arabic to Roman numeral.

Line 194-201: Please specify that you were speaking about average mg/100g of TVBN over the 21 days for each group.

Figure 2: Please separate the chicken-staph-citrox into 1% and 2% and include it in the figure 1. Please change the days on the figure 1 from Arabic to Roman numeral. Please include in the text that the TVBN in all the group increased over the study period but chicken-staph-citrox did not exceed 60mg/100g over the study period.

Figure 3: change the days from Arabic o Roman numeral.

Author Response

Dear Editor of Animals Journal

Thank you for your advices about our manuscript Title:  Effects of Citrox treatment on the survival of 3 methicillin-resistant Staphylococcus aureus 4 (MRSA) in chicken fillets packed under vacuum

Manuscript ID : animals-593597

The following are the modifications suggested by the two Reviewers point by point

Open Review

English language and style

( ) Extensive editing of English language and style required 
(x) Moderate English changes required 
( ) English language and style are fine/minor spell check required 
( ) I don't feel qualified to judge about the English language and style 

Yes

Can be improved

Must be improved

Not applicable

Does the introduction provide sufficient background and include all relevant references?

(x)

( )

( )

( )

Is the research design appropriate?

(x)

( )

( )

( )

Are the methods adequately described?

(x)

( )

( )

( )

Are the results clearly presented?

(x)

( )

( )

( )

Are the conclusions supported by the results?

(x)

( )

( )

( )

Comments and Suggestions for Authors

Reviewer 1

Line 2-3 About the title I suggest to change to the following: Citrox improves the quality and shelf life of chicken fillets packed under vacuum and protects against some foodborne pathogens

 Because through doing Similarity found high ratio of similarity near our title

Line 14-15: The samples were treated with 106 cfu/g of meat, vacuum-packed (VP), and 4°C. This statement is not clear, please rewrite this. What sample are you talking about here? Citrox or what? If its citrox, then be specific. Also, change “and 4oC” to at 4oC.

Line 17-19, the statement was rewritten again

Line 17: …during the storage period of meat samples.

Line 20, your suggestion was added

Line 17-24: decreased in growth of MRSA at what concentration of citrox? Please specified.

Line 22 was modified decreased after treatment with 2% cirox…………………….

Line 83-86: Please specify the sources of the citrox contents (manufacturers)

Line 88 citrox solution was prepared in our laboratory from pure chemical

Line 88: five hundred and twelve

Line 93 was changed to Five hundred and twelve

149-158: I have concern with this sensory test, apart from the fact that the cooked chicken meat is contaminated with MRSA, there was no information on the blinding of the panel of food scientists tasting the meat. How are we sure these 5 persons are not lab members that are familiar with the whole study and experiment. I am not quite sure of the validity of the result and inference from this section. Were the 5 persons blinded to the treatment groups?

Yes, the 5 person from the Department not from the laboratory and just they responsible and have the enough experience in food science to evaluate the different treatments. Many of our work here dependent on sensory analysis, and these persons specify and always done as this work.     

Line 160-164: Write something like Analysis of Variance (ANOVA) was used to determine the difference between the groups and periods of storage in a completely randomized factorial design. Completely randomized factorial design is not a statistical test but a study design.

Line 169-171 was changed as your suggestion

Result and Discussion

Line 173: were the chicken injected with MRSA or the meat samples treated in MRSA solution?

Line 184 the statement changed to the meat sample treated with MRSA solution

You are muddling things up here, be explicit as much as possible. State your result succinctly. You are moving from one result to another for example:

You have to describe changes/growth seen at each days of observation. You should move from (line 175) day 3 and 9 for positive control to day 12 of negative control and day 21 (line 177) of group C and D (line 179). You have to be explicit enough.

Line 185- 191 I was rewrite it again and moved as your suggestion

Line 181-183: at what day of study or were you talking about average reduction across the study days?

Line 193, ……………….on the 9th day

Line 186: delete Puhakka, 2010

Line 197 Puhakka. 2010 was deleted

Line 184-186: You did not measure temperature in your study, why talking about temperature here? Also the sentence does not connect with preceding and subsequent sentences.

I meant the temperature of storage 4 ºC, because some authors mentioned other temperatures for storage.

Line 186-191: Please rewrite this lines, it does not make sense, the sentences are all over the place and does not marry together

Yes, you are right it is not clear so, I was deleted the following paragraph from Line 199-203.

(There are many objective tests that suggest to prove meat spoilage. According to the standardization organization for the G.C.C. (GSO), 2007, the maximum allowable limit of microbial counts is 107 log cfu/g. Similarly, the permissible limit of TVC set by the ICMSF, 1986 is log 107 (for raw chicken, fresh or frozen).

Figure 1: Please separate the chicken-staph-citrox into 1% and 2% and include it in the figure 1. Why was the TVC for the chicken-staph increased and then declined to almost the level of chicken-staph citrox by day 21? Can you say something about this in the result and discussion? Also, please change the days on the figure 1 from Arabic to Roman numeral.

Figure 1.  I was separated the chicken staph citrox into 1 and 2% and included in the figure,

The days was changed to Roman numeral.

Line 194-201: Please specify that you were speaking about average mg/100g of TVBN over the 21 days for each group.

Line 211-216 was

Figure 2: Please separate the chicken-staph-citrox into 1% and 2% and include it in the figure 1. Please change the days on the figure 1 from Arabic to Roman numeral. Please include in the text that the TVBN in all the group increased over the study period but chicken-staph-citrox did not exceed 60mg/100g over the study period.

Figure 2. citrox 1% and 2% were separated as figure 1 and Days were changed to Roman Numerical, the statements you suggested was added Line 211-212

AND Your suggestion about The TVBN in all the groups increased over the study period ………..was added Line 211-213.

Figure 3: change the days from Arabic o Roman numeral.

Figure 3. the days were changed to Roman numerical.

Reviewer 2 Report

AAs spiker meat with S aureus but then only provide data for TVC. this is main flaw of the paper.

Author Response

The data of the paper studied the effect of citrox at different concentration 1 and 2% on the growth, total count of a pathogenic bacteria methicilline resistant Stahylococcus aureus  distributed in chicken meat, many researcher did not expose to microbe. How can limit the number of it and we succeded  to extend the shelf life more than 21 days at 4 degree centigradeand the proof is  TVB  ratio in coparison with other treatments.

English editing was done via Nature Springer English Editing
